# Improving Deep Learning by Inverse Square Root Linear Units (ISRLUs)

## Abstract

We introduce the "inverse square root linear unit" (ISRLU) to speed up learning in deep neural networks. ISRLU has better performance than ELU but has many of the same benefits. ISRLU and ELU have similar curves and characteristics. Both have negative values, allowing them to push mean unit activation closer to zero, and bring the normal gradient closer to the unit natural gradient, ensuring a noise-robust deactivation state, lessening the over fitting risk. The significant performance advantage of ISRLU on traditional CPUs also carry over to more efficient HW implementations on HW/SW codesign for CNNs/RNNs. In experiments with TensorFlow, ISRLU leads to faster learning and better generalization than ReLU on CNNs. This work also suggests a computationally efficient variant called the "inverse square root unit" (ISRU) which can be used for RNNs. Many RNNs use either long short-term memory (LSTM) and gated recurrent units (GRU) which are implemented with $\tanh$ and sigmoid activation functions. ISRU has less computational complexity but still has a similar curve to $\tanh$ and sigmoid.

## 1 Introduction

Two popular activation functions for neural networks are the rectified linear unit (ReLU) (Glorot et al., 2011) and the exponential linear unit (ELU) (Clevert et al., 2015). The ReLU activation function is the identity for positive arguments and zero otherwise. The ELU activation function is the identity for positive arguments and has an exponential asymptotic approach to -1 for negative values.

From previous analysis of the Fisher optimal learning, i.e., the natural gradient (Amari, 1998; Clevert et al., 2015), we can reduce the undesired bias shift effect without the natural gradient, either by centering the activation of incoming units at zero or by using activation functions with negative values. We introduce the inverse square root linear unit (ISRLU), an activation function like ELU, that has smoothly saturating negative values for negative arguments, and the identity for positive arguments. In addition this activation function can be more efficiently implemented than ELU in a variety of software or purpose-built hardware.

## 2 Inverse Square Root Linear Unit (ISRLU)

The *inverse square root linear unit* (ISRLU) with $\alpha$ is

$$f(x) = \begin{cases} x & \text{if } x \geq 0 \\ x\left(\frac{1}{\sqrt{1+\alpha x^2}}\right) & \text{if } x < 0 \end{cases}, \qquad f'(x) = \begin{cases} 1 & \text{if } x \geq 0 \\ \left(\frac{1}{\sqrt{1+\alpha x^2}}\right)^3 & \text{if } x < 0 \end{cases} \tag{1}$$

The ISRLU hyperparameter $\alpha$ controls the value to which an ISRLU saturates for negative inputs (see Fig. 1). ISRLUs and ELUs have very similar curves so at a high level one would expect to see the same general characteristics in most cases. ISRLUs have smooth and continuous first and second derivatives. ELUs are only continuous in the first derivative (see Fig. 1). In contrast, ReLU is non-differentiable at zero. Since ISRLUs and ELUs share most of the same characteristics we use the same weight initialization guidelines as are used for ELUs (Clevert et al., 2015)).

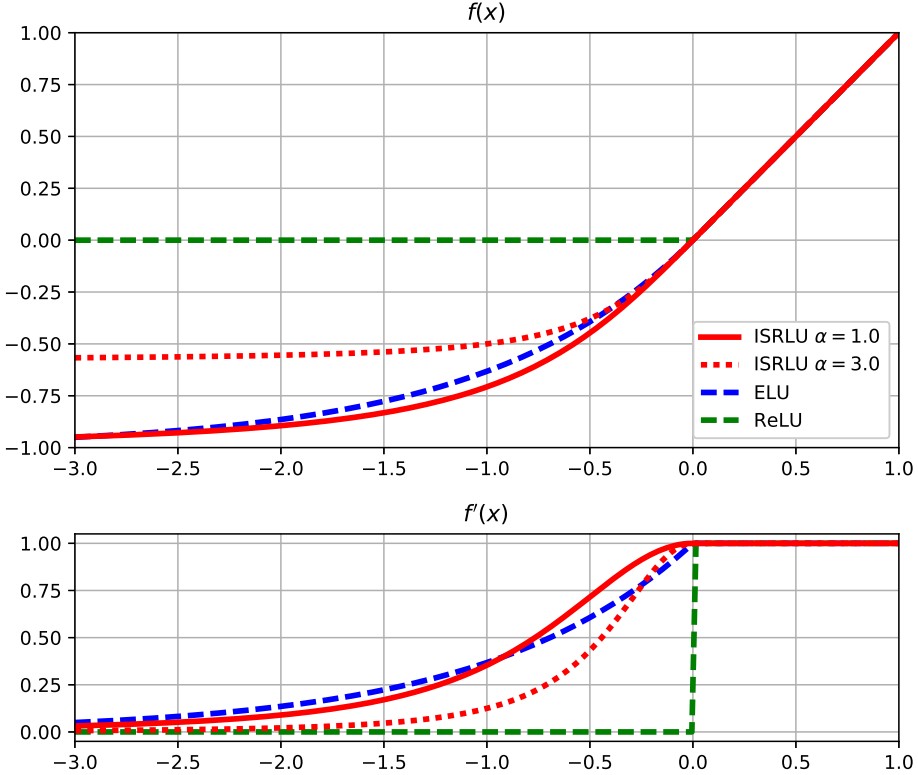

Figure 1: The inverse square root linear unit (ISRLU), ISRLU ($\alpha = 1; \alpha = 3$), ELU ($\alpha = 1$), and ReLU; and their first derivatives.

The primary advantage of ISRLU is in its reduced computational complexity compared to ELU. Inverse square roots are faster to calculate than exponentials. When calculating ISRLU for negative inputs, first one calculates $1/\sqrt{1 + \alpha x^2}$. Multiplying this function by $x$ provides the value for the forward calculation. Multiplying this function by itself twice (i.e. cubing) provides the value for back-propagation.

With $\alpha = 1$, ISRLU saturation approaches -1. With $\alpha = 3$, the negative saturation is reduced, so a smaller portion of the back-propagated error signal will pass to the next layer. This allows the network to output sparse activations while preserving its ability to reactivate dead neurons. Note that under variations of the $\alpha$ parameter, the ISRLU curve and its derivative remain smooth and continuous. Future work will establish what deeper saturation ($\alpha < 1$) is appropriate when applying ISRLU to self-normalizing neural networks (Klambauer et al., 2017).

In the same manner as parametric ReLUs (PReLUs) only one additional hyperparameter is required and methods can be used to directly learn its value during back-propagation (He et al., 2015). Similarly, ISRLU's $\alpha$ can be learned during the training phase along with the weights and biases. Indeed for PReLUs, He et al. (2015) have empirically shown that learning the slope parameter "a" gives better performance than manually setting it to a pre-defined value.

## 3 ACTIVATION FUNCTION PERFORMANCE

Shah et al. (2016) showed that ELU was faster than the combination of ReLU and Batch Normalization for deep neural network (DNN) ResNet architectures. On CIFAR-10 and CIFAR-100 they showed that ELU not only speeds up learning but also improves the accuracy as the depth of the convolutional neural network (CNN) increases.

More than learning rate needs to be considered when evaluating the overall performance of CNNs. The amount of time and computational resources required to perform both the convolutions and activation functions combined should be considered.

The trend in CNNs is that less time is being spent calculating convolutions. There are three factors that we are seeing. First is that small convolution filters such as 5x5 or 3x3 filters are the basis of many architectures. Second, architectures as Inception-v3 and Inception-v4 now decompose 2d filters such as a 3x3 into a 3x1 filter and a 1x3 filter (Szegedy et al., 2016). Third, more efficient calculations of convolution that rely on techniques such as Winograd's minimal filtering algorithm (Lavin & Gray, 2016; Winograd, 1980) are being used for 3x3 and smaller filters as are FFTs to reduce calculation time in 5x5 or larger filters. All of these techniques reduce the amount of calculations for each element in the convolution output.

Table 1 shows "cycles per output element" for an Intel Xeon Platinum 8160 (Skylake).

Table 1: Computational complexity of various filter sizes

| Convolution | FP Multiplies | FP Adds | Cycles per output element (CPE) |
|---|---|---|---|
| 5x5 | 25 | 24 | $\sim$4.25 |
| 3x3 | 9 | 8 | $\sim$1.53 |
| 3x1, 1x3 Inception-v3, -v4 | 3 | 2 | $\sim$0.51 |

Due to all of these reductions in convolution computational complexity, activation function performance is now a greater part of overall learning performance.

Another characteristic that is changing with the use of smaller filters is the decrease in the compute intensity (Carlile, 1993a;b), which raises the importance of memory systems performance for CNNs. The compute intensity of an algorithm is the ratio of the number of operations divided by number of words accessed. For a given algorithm it is straightforward to calculate the upper bound of the computation rate that can be supported on a given memory bandwidth.

## 3.1 ACTIVATION FUNCTION IMPLEMENTATION

The main advantage of ISRLU over ELU is that it is based on the inverse square root, which has been faster to evaluate than the exponential for many generations of systems. In the past, whenever it has not been faster, optimization potentials for inverse square root implementation improvement have been found. It is instructive to understand the current CPU performance of the inverse square root intrinsic performance compared to exponentials and `tanh`.

Intel x86 CPUs with SIMD instructions have vector intrinsic functions to accelerate performance. Intel publishes CPE (Clocks per Element) for various vector functions on their "Vector Mathematics (VM) Performance and Accuracy Data" website, see Table 2 (Intel, 2017).

Table 2: CPU performance on vector inverse square root, Exp, Tanh (x86).

| Vector Function Single Precision (EP) | Intel Xeon E5-2699 v3 (Haswell AVX2) | Intel Xeon E5-2699 v4 (Broadwell AVX2) | Intel Xeon Platinum 8180 (Skylake AVX-512) |
|---|---|---|---|
| InvSqrt | 0.66 | 0.64 | 0.24 |
| Exp | 0.81 | 0.89 | 0.52 |
| Tanh | 4.19 | 4.43 | 0.78 |
| Exp/InvSqrt | 1.2$\times$ | 1.4$\times$ | 2.2$\times$ |
| Tanh/InvSqrt | 6.3$\times$ | 6.9$\times$ | 3.3$\times$ |

For example, on a 3x1 filter using ELU in the negative region, approximately the same CPE is required to evaluate the convolution as is required for the exponential (cf. Table 1 and Table 2). Improvements in activation function performance will impact overall time spent in each learning step.

We measured the vector performance of AVX2 implementations for the various activation functions. The dataset used was 50% negative and 50% positive. Results are shown in Table 3.

Table 3: Vector ISRLU, ISRU, ELU, and ReLU performance on AVX2 (Intel Core i7-7700 Processor [3.60 GHz "Kaby Lake"] ).

| Activation Function Single Precision | nsec/ element | ISRLU Perf Advantage | ISRLU (approx.) Perf Advantage |
|---|---|---|---|
| ReLU | 0.340 | 0.62× | 0.99× |
| ISRU (approx.) | 0.334 | 0.61× | 0.97× |
| ISRLU (approx.) | 0.344 | 0.62× | 1.00× |
| ISRLU | 0.551 | 1.00× | 1.60× |
| ELU | 1.447 | 2.63× | 4.21× |

These results show that ISRLU ($\alpha = 1.0$) is 2.6× faster than ELU. The fast approximation of ISRLU is within 1% of the evaluation speed of ReLU while still retaining all of the desired learning curve properties mentioned in this paper. This fast approximation for ISRLU on this processor has only $3 \times 10^{-4}$ maximum relative error (~11.6 accurate bits). One Newton-Raphson iteration doubles that to ~23.4 accurate bits out of the 24 bits of mantissa, and two iterations achieves full precision. We plan to evaluate if the fast approximation has similar learning rates of the full precision ISRLU.

## 3.2 A PRACTICAL TRICK FOR INVERSE SQUARE ROOT CALCULATION

It is instructive to look at a practical trick for the computation of the inverse square root as it may serve as inspiration for those implementing ISRLU in hardware. Software implementations on CPUs can take advantage of floating-point formats for faster evaluation of the inverse square root. John Carmack and Terje Mathisen are often associated with implementing fast inverse square root in 2002 (Lomont, 2003). In 1986, one of the authors of this paper originally invented this method, which was called "The K Method," to implement vector square root for the production FPS T Series Hypercube Supercomputer (Gustafson, 1986). William Kahan and K.C. Ng at Berkeley also independently discovered this around 1986.

Carmack & Mathisen only used one iteration of the Newton method after their fast approximation. One iteration had an error of approximately 0.175%, which was suitable for their graphics applications. Since various piecewise functions have been used to approximate activation functions for CNNs and RNNs, part of our future research will look into if fast approximations to ISRLUs are suitable for DNNs.

Another avenue to look at for hardware implementations of the inverse square root is table-lookup hardware. Our expectation is that an efficient hardware approximation for the inverse square root should take about the same execution time as a fused multiply and add (FMA).

## 4 EXPERIMENTS USING ISRLUS

We used TensorFlow (Abadi et al., 2016) to train a CNN on the (Lecun) MNIST dataset. We tested the MNIST gray images in 10 classes, 60k train and 10k test.

The first CNN architecture (see Table 4) in our experiments used 28x28 input, a convolutional layer with 6x6 with 6 feature maps, a convolutional layer with 5x5 with 12 feature maps, a convolutional layer with 4x4 with 24 feature maps, a fully connected layer of 1176 hidden units, and a softmax output layer with 10 units. Only a full-precision ISRLU was used in these initial tests due to time constraints.

Convolutional neural networks with ISRLUs ($\alpha = 1.0$, $\alpha = 3.0$), ELUs ($\alpha = 1.0$), and ReLUs were trained on the MNIST digit classification dataset while each hidden units activation was tracked. Each network was trained for 17 epochs by using ADAM optimizer with learning rate 0.003 exponentially decreasing to 0.0001 and mini-batches of size 100. The weights have been initialized to truncated normal with standard deviation 0.1. The training error of ISRLU networks decreases much more rapidly than for the other networks. We also calculated the final cross-entropy loss function for each test.

Table 4: Architecture 1 on MNIST with test accuracy and cross-entropy loss with different activation functions.

| Activation Function | DropOut pkeep | Max Test Accuracy | Cross-Entropy Loss |
|---|---|---|---|
| ISRLU $\alpha = 3.0$ | 0.25 | 99.30 | 2.308 |
| ELU | 0.40 | 99.29 | 2.395 |
| ISRLU $\alpha = 3.0$ | 0.40 | 99.27 | 2.530 |
| ReLU | 0.40 | 99.22 | 2.644 |
| ISRLU $\alpha = 1.0$ | 0.40 | 99.20 | 2.785 |
| ReLU | 0.25 | 99.17 | 2.798 |
| ELU | 0.25 | 99.09 | 2.892 |
| ISRLU $\alpha = 1.0$ | 0.25 | 99.00 | 3.124 |

The second CNN architecture (see Table 5) in our experiments used 28x28 input, a convolutional layer with 3x3 with 64 feature maps, a convolutional layer with 3x3 with 64 feature maps, 2x2 Maxpooling, DropOut, a convolutional layer with 3x3 with 64 feature maps, a convolutional layer with 3x3 with 64 feature maps, 2x2 Maxpooling, DropOut, a fully connected (FC) layer of 512 hidden units, and a softmax output layer with 10 units. Full-precision ISRLU was used.

Convolutional neural networks with ISRLUs ($\alpha = 1.0$, $\alpha = 3.0$) and ELUs ($\alpha = 1.0$) were trained on the MNIST digit classification dataset while each hidden units activation was tracked. The network was trained for 20 epochs by using ADAM optimizer with learning rate 0.003 exponentially decreasing to 0.0001 and mini-batches of size 100. The weights have been initialized to truncated normal with standard deviation 0.1.

Table 5: Architecture 2 on MNIST with test accuracy and cross-entropy loss with different activation functions.

| Activation Function | DropOut pkeep | Max Test Accuracy | Cross-Entropy Loss |
|---|---|---|---|
| ISRLU $\alpha = 1.0$ | 0.7 conv 0.4 FC | 99.32 | 2.334 |
| ISRLU $\alpha = 3.0$ | 0.7 conv 0.4 FC | 99.30 | 2.389 |
| ELU | 0.7 conv 0.4 FC | 99.29 | 2.225 |

We did not expect significant differences in accuracy in ISRLU and ELU in this test of shallow networks due to the similar nature of the curves. The cross-entropy loss was reasonable, at between 2 and 3.2 for all activation functions. Future testing will be done on deeper networks where we expect larger advantages that are similar to ELU (Clevert et al., 2015; Shah et al., 2016).

## 5 INVERSE SQUARE ROOT UNIT (ISRU)

The work with ISRLU in this paper suggests that the *inverse square root unit* (ISRU) may be useful for a variety of neural networks. ISRUs are defined as:

$$f(x) = x \left( \frac{1}{\sqrt{1 + \alpha x^2}} \right), \qquad f'(x) = \left( \frac{1}{\sqrt{1 + \alpha x^2}} \right)^3 \tag{2}$$

In RNNs that use LSTM (Hochreiter & Schmidhuber, 1997) and GRU (Chung et al., 2014), the most common activation functions are sigmoid and `tanh`. We assert that ISRUs can be more efficient calculation than `tanh` and be more efficient than sigmoid when properly shifted and scaled. As shown above in Table 2, the inverse square root is 3x to 6x faster than `tanh` (depending on x86 architecture). ISRUs will be an area of our future research.

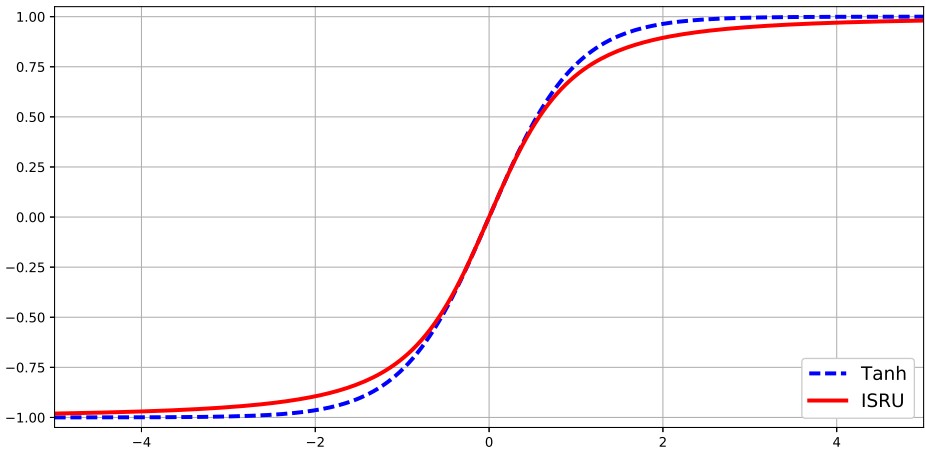

Figure 2: The inverse square root unit (ISRU) and `tanh` functions.

## 6 CONCLUSION

Activation function performance is becoming more important overall in convolutional neural networks (CNNs) because of the trending reductions in the computational complexity of the convolutions used in CNNs. We have introduced a new activation function, the inverse square root linear unit (ISRLU) for faster and precise learning in deep convolutional neural networks. ISRLUs have similar activation curves to ELUs, including the negative values. This decreases the forward propagated variation and brings the mean activations to zero. Mean activations close to zero decreases the bias shift for units in the next layer which speeds up learning by bringing the natural gradient closer to the unit natural gradient. Future work may prove the effectiveness of applying ISRLUs and the related ISRUs to other network architectures, such as recurrent neural networks, and to other tasks, such as object detection. ISRLUs have lower computational complexity than ELUs. Even greater savings on computation can be realized by implementing ISRLUs in custom hardware implementations. We expect ISRLU activations to increase the training efficiency of convolutional networks.

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
