# OpenReview forum: "Improving Deep Learning by Inverse Square Root Linear Units (ISRLUs)"
_ICLR.cc/2018/Conference — Reject_

### Official Review · AnonReviewer1 · 2017-11-27
**This paper introduces a new nonlinear activation function for  neural networks, i.e., Inverse Square Root Linear Units (ISRLU).**

**Rating:** 4
**Confidence:** 4

**Review:**

This paper introduces a new nonlinear activation function for  neural networks, i.e., Inverse Square Root Linear Units (ISRLU). Experiments show that ISRLU is promising compared to competitors like ReLU and ELU.

Pros:
(1) The paper is clearly written.

(2) The proposed ISRLU function has similar curves with ELU and has a learnable parameter \alpha (although only fixed value is used in the experiments) to control the negative saturation zone.

Cons:
(1) Authors claim that ISRLU is faster than ELU, while still achieves ELU’s performance. However, they only show the reduction of computation complexity for convolution, and speed comparison between ReLU, ISRLU and ELU on high-end CPU. As far as I know, even though modern CNNs have reduced convolution’s computation complexity, the computation cost of activation function is still only a very small part (less than 1%) in the overall running time of training/inference.

(2) Authors only experimented with two very simple CNN architectures and with three different nonlinear activation functions, i.e., ISRLU/ELU/ReLU and showed their accuracies on MNIST. They did not provide the comparison of running time which I believe is important here as the efficiency is emphasized a lot throughout the paper.

(3) For ISRLU of CNN, experiments on larger scale dataset such as CIFAR or ImageNet would be more convincing. Moreover, authors also propose ISRU which is similar to tanh for RNN, but do not provide any experimental results.

Overall, I think the current version of the paper is not ready for ICLR conference. As I suggested above, authors need more experiments to show the effectiveness of their approach.

---

### Official Review · AnonReviewer2 · 2017-11-27
**Good design of the activation function, but needs more convincing results to show a win over ReLU.**

**Rating:** 5
**Confidence:** 4

**Review:**


Summary:
- The paper proposes a new activation function that looks similar to ELU but much cheaper by using the inverse square root function.

Contributions:
- The paper proposes a cheaper activation and validates it with an MNIST experiment. The paper also shows major speedup compared to ELU and TANH (unit-wise speedup).

Pros:
- The proposed function has similar behavior as ELU but 4x cheaper.
- The authors also refer us to faster ways to compute square root functions numerically, which can be of general interests to the community for efficient network designs in the future.
- The paper is clearly written and key contributions are well present.

Cons:
- Clearly, the proposed function is not faster than ReLU. In the introduction, the authors explain the motivation that ReLU needs centered activation (such as BN). But the authors also need to justify that ISRLU (or ELU) doesn’t need BN. In fact, in a recent study of ELU-ResNet (Shah et al., 2016) finds that ELU without BN leads to gradient explosion. To my knowledge, BN (at least in training time) is much more expensive than the activation function itself, so the speedup get from ISRLU may be killed by using BN in deeper networks on larger benchmarks. At inference time, all of ReLU, ELU, and ISRLU can fuse BN weights into convolution weights, so again ISRLU will not be faster than ReLU. The core question here is, whether the smoothness and centered zero property of ELU can buy us any win, compared to ReLU? I couldn’t find it based on the results presented here.
- The authors need to validate on larger datasets (e.g. CIFAR, if not ImageNet) so that their proposed methods can be widely adopted.
- The speedup is only measured on CPU. For practical usage, especially in computer vision, GPU speedup is needed to show an impact.

Conclusion:
- Based on the comments above, I recommend weak reject.

References:
- Shah, A., Shinde, S., Kadam, E., Shah, H., Shingade, S.. Deep Residual Networks with Exponential Linear Unit. In Proceedings of the Third International Symposium on Computer Vision and the Internet (VisionNet'16).

---

> ### Author Response · Authors · 2017-12-04
> **Question on Shah reference**
>
> Many thanks for your comments & observations on our paper.
>
> We referenced the Shah et al "Deep Residual Networks with Exponential Linear Unit" paper in our paper.
>
> You mention, "ELU without BN leads to gradient explosion"  But the paper you referenced seems to state they use ELU _without_ batch normalization (BN) and compared it favorably to ReLU+BN.
>
> Shah et al in the intro: "In this paper, we propose the use of exponential linear unit instead of the combination of ReLU and Batch Normalization in Residual Networks.  We show that this not only speeds up learning in Residual Networks but also improves the accuracy as the depth increases."
>
> We're a bit confused about your statement... can you clarify?
>
> BTW, all of our experiments with Mnist didn't use BN.  In addition we are finishing up what look like favorable results for ISRLU (without BN) on CIFAR, GANs, and CapsuleNets.  We would like to add these experiments to our paper to broaden our test cases.

---

### Official Review · AnonReviewer3 · 2017-11-27
**Strong lack of results supporting significance**

**Rating:** 3
**Confidence:** 4

**Review:**

Summary:
The contribution of this paper is an alternative activation function which is faster to compute than the Exponential Linear Unit, yet has similar characteristics.
The paper first presents the mathematical form of the proposed activation function (ISRLU), and then shows the similarities to ELU graphically. It then argues that speeding up the activation function may be important since the convolution operations in CNNs are becoming heavily optimized and may form a lesser fraction of the overall computation. The ISRLU is then reported to be 2.6x faster compared to ELU using AVX2 instructions. The possibility of computing a faster approximation of ISRLU is also mentioned.
Preliminary experimental results are reported which demonstrate that ISRLU can perform similar to ELU.

Quality and significance:
The paper proposes an interesting direction for optimizing the computational cost of training and inference using neural networks. However, on one hand the contribution is rather narrow, and on the other the results presented do not clearly show that the contribution is of significance in practice.
The paper does not present clear benchmarks showing a) what is the fraction of CPU cycles spent in evaluating the activation function in any reasonably practical neural network, b) and what is the percentage of cycles saved by employing the ISRLU.
The presented results using small networks on the MNIST dataset only show that networks with ISRLU can perform similar to those with other activation functions, but not the speed advantages of ISRLU.
The effect of using the faster approximation on performance also remains to be investigated.

Clarity:
The content of the paper is unclear in certain areas.
- It is not clear what Table 2 is showing. What is "performance" measured in? In general the Table captions need to be clearer and more descriptive. The acronym pkeep in later Tables should be clarified.
- Why is the final Cross-Entropy Loss so high even though the accuracy is >99% for the MNIST experiments? It looks like the loss at initialization was reported instead?

---

### Public Comment · (anonymous) · 2017-10-30
**Fast, and compact approximation of the exponential function**

Schraudolph introduced [1,2] and Cawley later revised [3] the "Fast, and compact approximation of the exponential function".
I'm wondering how the speed comparisons would change if the exponentials in the sigmoid and ELU activations are replaced accordingly.

[1] http://www.mitpressjournals.org/doi/10.1162/089976699300016467
[2] https://nic.schraudolph.org/pubs/Schraudolph99.pdf
[3] http://www.mitpressjournals.org/doi/abs/10.1162/089976600300015033
[4} https://martin.ankerl.com/2007/02/11/optimized-exponential-functions-for-java/

---

> ### Author Response · Authors · 2017-11-01
> **Response to Fast, and compact approximation of the exponential function**
>
> Yes, we have seen this method that shares the ideas of the "K method" that were decades years ago for inverse square root.  These can improve performance for Exp.  The implementations you pointed at were 64-bit approximations. While most DNN is done in 32-bit, 16-bit, etc.  They took a few other optimization that you pointed to such as no bounds checking (which is probably ok for a well-designed AI implementation.  Also these were scalar implementations) and they not vector implementations which depending on the hardware may  bring up issues.
>
> Another way to get faster intrinsic performance is various lower-precision implementations. In fact. we've even coded up some of our low lower-precisions intrinsics ourselves. For some background we'd suggest the classic: J.Hart, E.W. Cheney, et al, Computer Approximations, Publisher: Krieger Pub Co (July 1, 1978), ISBN-10: 0882756427 Side note: you should "reshoot" your own Chebyshev coefficients using Remez (not using the co-efficents in the book)
>
> But the bottom line is that inverse-square root, as mentioned in the paper has been faster than exp.
>
> With an approximation you can then basically double the number of bits of accuracy with each Newton-Raphson iteration, which is easy to vectorize and does not increase the flop count too much.
>
> The other point we made in the paper is that ISRLU has a very natural way to introduce an alpha that is smooth and continuous for 1st/2nd derivatives.  Yes there are other ways of introducing alpha into ELU functions (https://arxiv.org/abs/1704.07483) but we still find ISRLU more natural.
>
> Thanks for you comment, we hope that ISRLU & ISRU be considered especially for purpose-built AI DNN hardware.

---

> > ### Author Response · Authors · 2017-11-07
> > **inverse square root should always be faster than exp**
> >
> > With our experience on a wide variety of architectures and implementations of instrinsics, if inverse square root is not faster than exp one should look closely at the hardware/software implementation as this is a clue that inverse square root can be better implemented.

---

### Decision · Program_Chairs · 2018-01-29
**ICLR 2018 Conference Acceptance Decision**

**Decision:**

Reject

**Comment:**

The authors introduce a new activation function which is similar in shape to ELU, but is faster to compute.   The reviewers consider this to not be a significant innovation because the amount of time spent in computing the activation function is small compared to other neural network operations.